# ‘I Was Smoking a Lot More during Lockdown Because I Can’: A Qualitative Study of How UK Smokers Responded to the Covid-19 Lockdown

**DOI:** 10.3390/ijerph18115816

**Published:** 2021-05-28

**Authors:** Rachel O’Donnell, Douglas Eadie, Martine Stead, Ruaraidh Dobson, Sean Semple

**Affiliations:** Institute for Social Marketing and Health, Faculty of Health Sciences and Sport, University of Stirling, Stirling FK9 4LA, UK; douglas.eadie@stir.ac.uk (D.E.); martine.stead@stir.ac.uk (M.S.); r.p.dobson@stir.ac.uk (R.D.); sean.semple@stir.ac.uk (S.S.)

**Keywords:** Covid-19, lockdown, smoking, home, second-hand smoke, qualitative

## Abstract

This study explored how Covid-19 lockdown restrictions affected people’s daily smoking routines and behaviours, including adherence and modifications to pre-established smoking restrictions in the home. Semi-structured telephone interviews were conducted with smokers and non-smokers from smoking households 19 to 27 weeks after the first full UK lockdown ended in May 2020. A non-probability purposive sample representing 25 adults aged 21 or over living in households with at least 1 smoker were recruited to the study. A quota sampling strategy was used, according to age, gender, smoking status, family status, household composition, householder access to outdoor space, and change to work-life status. Most participants found lockdown increased the amount of time spent at home, where stresses associated with confinement, curtailment of social routines, removal of barriers and distractions to smoking due to home working, and feelings of boredom all contributed to increased smoking. Fewer factors were identified as reducing smoking during lockdown. Prominent examples included disruption to habitual smoking patterns and distraction from smoking associated with spending more time doing outdoor activities. Pressures placed on physical space and lack of privacy due to the confinement at home were responsible for displacement of smoking within the home, leading to breaking of smoke-free rules and family tensions, and in some cases to greater awareness amongst parents that their children smoked. Changes in daily routines associated with lockdown affected and displaced smoking behaviour both positively and negatively. Health improvement interventions could seek to harness positive changes in smoking associated with any future lockdown approaches. New home-working norms highlight the need for employers to support staff to reduce their smoking and to remain smoke-free.

## 1. Introduction

Many governments worldwide responded to the Covid-19 pandemic by introducing lockdowns restricting social contact to limit the transmission of the disease and reduce the risk of health services being overwhelmed. Lockdowns were accompanied by Government communications, encouraging the public to stay at home except for essential purposes. It is important to examine the public health impacts of attempting to contain the spread of Covid-19, as well as the impacts of Covid-19 more generally, as lockdown has been described as a public health intervention in its own right [1].

From the early stages of the pandemic, evidence suggested that smoking is associated with increased severity of disease and death in hospitalised Covid-19 patients [2,3]. Evidence in the UK and elsewhere also suggests that Covid-19 and associated lockdown measures have impacted on smoking behaviours in varied ways. Some smokers report increased daily tobacco use, suggesting that smoking has been used as a coping mechanism to deal with pandemic-related boredom, loneliness, anxiety, anger and/or stress [3,4,5,6,7,8]. No changes in self-reported smoking rates, and reduced smoking consumption have been reported in other cases [6,7,8]. Some studies have reported increases in cessation and quit attempts [9], and increased motivation to quit for some study participants [10], potentially reflecting concerns about contracting Covid-19 and becoming severely ill as a result [7,8]. A few studies have reported changes in tobacco purchasing patterns, with some cigarette smokers having large stocks of cigarettes at home to avoid leaving the house every day to purchase more [10,11].

Changes in smoking behaviours may result from pandemic-related changes to the structure of everyday lives [8,10]. Whilst several studies have focused on changes in smoking consumption, little is known about the impact of Covid-19 and lockdown restrictions on smoking behaviours in the home. It has been suggested that spending more time at home with children and/or non-smoking partners could lead to reduced smoking to protect family members from the harms of second-hand smoke (SHS), as familial connections are pivotal in shaping smoking behaviours [12]. Stay-at-home restrictions could lead to cessation in smokers who are unable or unwilling to smoke in the home, because of home smoking rules set by landlords or because they have children in the household [9]. One recent study conducted in Israel suggested that nearly one in five participants (19.5%) reported a change in home smoking rules during the pandemic. Two-thirds (12.9%) indicated a change to reflect less exposure to other household members, whilst one-third (6.6%) indicated a change to reflect increased exposure to other household members [13]. In contrast, findings of a large (n = 6003) cross-sectional survey conducted in Italy suggest that increased smoking during the first full lockdown was not accompanied by a relaxation of smoke-free home rules, with only a negligible proportion of the overall sample (0.11%) reporting an increase in SHS exposure at home [14]. However, UK survey data suggests that 12% of smokers living with children reported smoking indoors more than they did before the first lockdown [15].

The UK Government announced the first lockdown in the UK on 23 March 2020, imposed through public health legislation, with separate regulations made in the UK, Scottish and Welsh Parliaments and the Northern Ireland Assembly. England, Scotland, and Wales introduced lockdown restrictions from 26 March to 9 May 2020, and Northern Ireland from 28 March to 17 May 2020, with only minor differences in their respective approaches. The key public health message at this time was to stay at home with limited exceptions including basic food shopping, exercise once per day, medical need and travelling for work when absolutely necessary. Almost all businesses, facilities, places of worship and schools were closed during this time; schooling continued only for children of key workers and children considered vulnerable. One in eight households (12%) in Great Britain had no access to a private or shared garden, patio, or balcony during lockdown, rising to more than one in five in London (21%). Lockdown also highlighted inequalities in access to outdoor space. People of all ethnic minorities are less likely to have outdoor space at home than those of white ethnicity, and people in semi-skilled/unskilled manual occupations are almost three times as likely as those in managerial, administrative and professional occupations to be without a garden (21% v 7%) [16]. Previous research suggests lack of access to private outdoor space is a key barrier to creating/maintaining a smoke-free home [17]. On this basis, the aim of this study was to explore how Covid-19 lockdown ‘stay at home’ restrictions affected daily smoking routines and behaviours, and to investigate changes to pre-established smoking restrictions in the home.

## 2. Materials and Methods

### 2.1. Sample and Recruitment

A non-probability purposive sample was generated from an independent UK-wide market research panel, comprised of members of the public who consented to be contacted to participate in online research surveys and telephone interviews. All panel members aged 21 or over (11,847 panel members in total) were invited by email to take part in the study and complete a screening questionnaire. Based on screening questionnaire responses, to avoid factors that might confound the impact of lockdown on time-activity patterns at home, those who had (a) moved home between January 2020 and the start of lockdown, (b) been admitted to hospital with COVID-19, or (c) a household member admitted to hospital with COVID-19 were excluded from taking part. Those who opted not to confirm their age, and those who had taken part in three or more studies as part of the independent market research panel in the last six months were also excluded from participating. On this basis, 572 responders were identified as eligible to participate. Thirty-two participants were recruited in order of response to meet a quota sampling strategy based on seven sample variables: gender, age, smoking status, family status, household composition, household access to private outdoor space, and pandemic-related change to work-life status (see Appendix A for sampling plan). These variables were identified to explore differing impacts of lockdown restrictions on time spent inside the home and to include important determinants of household smoking behaviour. The numbers recruited into the study met the proportions established for each of the seven sample variables, with the exception of access to private outdoor space; those without access were slightly under-sampled. Seven of thirty-two participants lived in non-smoking households. This paper presents findings from all households with smokers (n = 25).

### 2.2. Data Collection

Semi-structured telephone interviews were conducted with adults living in smoking households across the UK between September and November 2020, 19 to 27 weeks after the first full lockdown. The interviews (conducted by DE and RO) used a schedule devised for the wider study, which sought to model the health consequences of changes in personal exposure to fine particulate air pollution (PM_2.5_) from SHS and other sources during the UK lockdown to reduce community transmission of Covid-19. Interview questions were developed by DE and RO in discussions with the wider team, and included a focus on changes in household smoking activity, changes in tobacco consumption, frequency of indoor smoking, and changes to daily routines associated with lockdown restrictions, including those related to the amount of time spent indoors at home alone and in the presence of smokers. This provided a detailed picture of how people spent their time before, during, and after full lockdown, and whether and how their smoking behaviour, and the smoking behaviour of others in the household, was affected by disruptions associated with lockdown. The interviews also explored ways in which lockdown restrictions affected household composition and social interactions between household members, including the impact of smoking on family dynamics and adherence and modifications to smoking restrictions in the home. Interview questions were reviewed by DE and RO after the first five interviews were conducted, with no adjustments required.

All participants were emailed a copy of the participant information sheet and consent form in advance of the interview and given an opportunity to ask questions before deciding whether to participate. Participants were offered a £30 e-voucher as a gesture of thanks for taking part. The interviews, which lasted up to 60 min, were digitally recorded with the participants’ consent. Audio-files were fully transcribed by a professional transcription agency and anonymised for reflexive thematic analysis. Ethical approval was provided by the General University Ethics Panel at the University of Stirling (GUEP 19 20 957R).

### 2.3. Data Analysis

Analysis was led by DE and supported by RO. A core set of themes based on the interview schedule were identified and refined by DE and RO and formed a framework for coding the transcripts (by DE) using NVivo 12 software. The transcripts underwent two stages of analysis. Firstly, they were organised using the thematic framework and potential themes were identified through a process of familiarisation with transcript texts. Then, the transcripts and coded data were re-read and analysed by DE and RO to build a series of individual narratives and case histories. At regular points during the process of data analysis, DE and RO met with members of the wider research team (MS and SS) to discuss identified codes and categories, the interpretation of data and potential new areas of enquiry. The analysis process therefore drew on the combined insights of the researchers most closely involved in data collection and analysis, with wider team perspectives and expertise associated with the topic area.

These analyses were also discussed regularly within the wider team, allowing the researchers to identify patterns and differences across the data as a whole and to develop separate participant behaviour profiles.

The study was reported in accordance with the 32-item checklist of Consolidated Criteria for Reporting Qualitative Research (COREQ) [18].

## 3. Results

Twenty-five adults aged twenty-one and over (average age 50 years, range 22–73 years) took part in the study. Two prospective participants were uncontactable at the time of the scheduled interview. Table 1 summarises participants’ socio-demographic characteristics, smoking status, types of disruption experienced to working lives, changes experienced in numbers of hours spent at home, and changes in smoking levels.

The sample included a broad spectrum of income levels as well as comparable numbers by gender (twelve women and thirteen men), household smoking status (twelve all-smoking households and thirteen mixed status households). Nine participants were non-smokers and sixteen smoked; four lived alone, nine lived in all-adult households and twelve lived in households with school age/younger children. Fourteen participants lived in homes with access to private outdoor space such as a garden, yard, or balcony, and eleven did not. Sixteen participants lived in homes with smoke-free rules, and nine lived in homes where smoking was permitted (in at least one room).

Lockdown affected participants’ working lives in a number of ways. Fifteen participants continued in employment during lockdown: two carried on working with little change; seven were required to work from home; and six were furloughed. One participant who was self-employed ceased working during lockdown unsupported, and two participants were unemployed, one of whom lost his job during full lockdown. Of the remaining seven participants, six were retired, and one was a full-time carer for his wife.

Lockdown had a significant impact on the amount of time spent at home and on levels of smoking. On average, the estimated amount of time spent at home across a 24-h day increased by 5.5 h (range −2.3–+11.3 h), with only one participant recording a decrease (P08), attributed to spending more time outdoors. Approximately three-quarters (n = 17) of participants described a change in their smoking levels during/following full lockdown: for most reporting a change, smoking increased (n = 11), with three of these participants reporting a subsequent attempt to reduce intake. Just under a quarter (n = 6) of participants reported reduced smoking during lockdown. In one case this reduction had not been maintained by the time of interviews. Some reductions in smoking were associated with the use of e-cigarettes as a substitute or partial replacement for tobacco.

From participants’ accounts of how the changes brought about by lockdown affected their own/other householders’ smoking behaviour, four common themes were generated: changes in routine and boredom; confinement and stress; smoking concealment; and diversionary activity. These four themes reflected both potential influencers on smoking behaviour and ways in which smoking behaviour changed. Illustrative verbatim quotes for each theme are provided alongside participant gender, age, smoking status and work-life status, and household composition (single person household, all adult household, household with children) and household access to private outdoor space. All data have been anonymised to protect participant identities.

### 3.1. Changes to Routine and Boredom

Changes to work and social routines due to lockdown were perceived to have an impact on smoking behaviours. Being furloughed, having a curtailed social life, and finding it difficult to establish new routines led to feelings of listlessness and boredom, which in some cases were contributory factors to increased smoking:

“*My smoking increased twofold… It was boredom more than anything, playing a game or watching TV, and then obviously it would go on a break or it would finish, so then I’d go for a fag*.”(P15, male, 31, furloughed, smoker, household with children, access to private outdoor space)

Interviewer: “*So, you said you smoked more during lockdown?*”Respondent: “*Yeah, no sauna, no steam, no arranging for tennis, no meeting up with my friends, all those things*.” (P25, female, 61, home working, smoker, all adult household, no access to private outdoor space)

However, for some habitual smokers, furlough helped to break work-related patterns that had previously helped to facilitate smoking habits:

“*He (husband) was definitely smoking less. He said that himself on a couple of occasions… because at work he’s sort of…he’s in and out of jobs so when he’s been on his way to one job he has a cigarette and then he goes in, comes out, has a cigarette on his way to the next job. Whereas at home he’s not doing that.*” (P20, female, 28, furloughed, non-smoker, household with children, no access to private outdoor space)

For those trying to quit smoking, the disruption to social life caused by lockdown removed important triggers for smoking relapse, including drinking socially:

“*I tried (giving up smoking) before but I’ve never really stuck at it. If I went out drinking, I’d always start back up. And then obviously…how it is at the minute with social distancing, we couldn’t go out together for a fag at work with friends. So yeah, there have been some benefits to lockdown.*” (P15, male, 31, furloughed, smoker, household with children, access to private outdoor space)

Working from home was also associated with increased smoking, and in some cases with the displacement of smoking into homes. These changes in smoking behaviours were attributed to the removal of features of the workplace that had previously prevented or distracted participants from smoking, including regulated breaks, smoke-free workplace rules, social interaction with work colleagues and more varied work patterns and tasks:

“*I was smoking a lot more during lockdown, because I can, because at work they don’t like it if you go out every ten minutes, it’s a bit rude. So at work I would, you know, I’d start work at eight, and I’d go out sort of twice in the morning and then once in the afternoon, whereas at home I could be lighting up a cigarette every ten minutes*.” (P19, female, 54, home working, smoker, single person household, no access to private outdoor space)

The disruption caused by lockdown led some participants to find new ways of socialising, such as meeting with friends remotely or outdoors. These adaptions were also described as providing new opportunities to smoke and, in some cases, facilitated increased smoking:

“*We were doing Zoom calls, because you couldn’t go anywhere,… so there’d be like six or seven of us in our own houses drinking and Zooming….)and) because I was in my house, I would smoke a lot more. Whereas if we were in the pub then I’d have to keep going out for a cigarette…. also the pubs used to shut at 12, whereas we’d be Zooming ‘til two, three and four in the morning.*” (P19, female, 54, home working, smoker, single person household, no access to private outdoor space)

There was also some evidence that lockdown and the threat of Covid-19 infection affected tobacco purchasing routines and choices. For example, one smoker’s decision to switch to e-cigarettes was, in part, guided by a wish to limit regular local shopping trips for tobacco to minimise non-essential contact:

“*I was vaping full-time (*during lockdown*)… It was basically to try and save going to the shop every day (*for cigarettes*) to minimise contact with people. I didn’t want to have to go to a shop all the time.*” (P10, female, 45, furloughed, smoker, household with children, no access to private outdoor space)

### 3.2. Confinement and Stress

Loss of freedom and the stress of being confined for long hours at home were described as contributing to increased smoking, particularly amongst those living with children and with limited access to outdoor space:

*Interviewer: “So you found it more stressful during lockdown, is that right?*”

*Respondent: “Yes, (I was probably smoking more…) because you’re cooped up, aren’t you? Basically, in the house rather than having flexibility and a bit of freedom to go out and go to places*.” (P13, male, 36, home working, smoker, household with children, access to private outdoor space)

For some parents, confinement proved particularly challenging, with pressures of home schooling and exposure to criticism from children regarding their smoking adding to stress levels. The difficulties of living in a flat with a young family also affected intentions to quit during lockdown and precipitated increased smoking for some:

“*At first I saw it as an opportunity to cut down, but then I think as the situation progressed and having the kids at home I felt more stressed which in turn made me want to smoke more.…They were asking for help with their school work and I’d tell them to hold on a minute—‘Oh, you’re always in the kitchen smoking’, I’d get comments like that from them*.” (P24, female, 33, home working, smoker, household with children, no access to private outdoor space)

In other cases, the pressures of looking after children at home with limited access to outdoor space were compounded by additional work-related stresses, which led not only to increased smoking, but also to relaxation of smoking restrictions in the home:

“*He* (partner) *smoked more because it became very stressful at work….It was the fact none of his* (residential) *clients could go out, so he was having to do more for them in the home. Obviously, they would get cranky and stuff because they weren’t going out, doing their usual activities, so he was coming home, moaning… obviously about how stressed he was, and then my daughter would go off on one (*become agitated*), so obviously it all adds up*.” (P17, female, 22, furloughed, non-smoker, household with children, no access to private outdoor space)

However, not all families found lockdown stressful. For example, one mother described furlough as ‘*really enjoyable’*, providing an opportunity for her and her husband to ‘*slow down’* and spend more time together with their young family (P20, female, furloughed, non-smoker, household with children, no access to private outdoor space). These positive experiences were associated with a reduction in smoking following disruption of her husband’s work-related smoking patterns.

### 3.3. Smoking Concealment

Lockdown was also associated with efforts to conceal smoking from family members. This was more evident in the accounts of those with limited access to outdoor space, where non-adherence to existing smoke-free home rules and attempts to conceal smoking indoors from other householders could lead to conflict and increased family tensions:

“*He* (husband) *would go to the bathroom and open the window and then I’d go crazy because of the smell* (of smoke)… *He tries to close the door and fob me off (*make excuses for being in there*).*”(P08, female, 45, home working, smoker, all adult household, access to private outdoor space).

Concealment behaviours were also occasionally described by smokers who had access to private outdoor spaces. For example, one recalled moving from one part of the garden to another more private part in order to avoid disapproval from non-smoking members of his household:

“*I was just a bit more conscious that the family were around, you know* (*wife and daughter both non-smokers*). *So, if we were doing things outside, I’d be tending to not smoke as much in their presence. Well they’re avid anti-smokers. So I’m a scourge…They’re constantly bickering, ‘cause of the (my) diabetes more than anything else.*” (P03, male, 71, retired, smoker, all adult household, access to private outdoor space)

These pressures on living in shared home spaces were sometimes exacerbated when other family members came to live in the household during lockdown, for example sons and daughters returning from university, who were critical of their parents smoking, or where individuals avoided smoking in outdoor public spaces, due to perceived stigma and social unacceptability:

“*I would never be one of those people that would go outside and walk down the street with a cigarette, I’m not one of those people. If I leave the house, I don’t smoke… I wouldn’t smoke in public.*” (P08, female, 45, home working, smoker, all adult household, access to private outdoor space)

Children also emerged as a common theme in concealment narratives, for example, parents who sought to hide their smoking from their younger children found this more difficult during lockdown. In one case, a mother who assumed the role of full-time parent during lockdown was unable to smoke in the family flat during the day when her partner was at work. An unintended, positive consequence of these constraints on her freedom to smoke was that it led to her reduced smoking consumption:

“*She wouldn’t* (smoke in the flat)…. *She’d wait until I got home from work…. Even if she really needed a cigarette, she’d have to wait till I got home…. she’s never smoked up here* (in the flat). *The kids don’t even know she smokes… she’s always hid it from them.*” (P05, male, 41, working, smoker, household with children, no access to private outdoor space)

Parents also reported lockdown having a significant impact on their teenage children who smoked, with under-age smokers finding it more difficult to conceal their smoking (and e-cigarette use) from parents. In one case, parents described how lockdown had been instrumental in revealing their teenage sons’ smoking habit and the heightened challenges of addressing this at this time:

“*Pre-lockdown we didn’t think he* (teenage son) *went anywhere near tobacco, but since lockdown he’s...well, I’d say he’s partially addicted to nicotine. He tries to hide it…. I mean, he’s got a cough, which is horrendous, which we keep telling him is due to his smoking. But as I said, he’s suffering with some* (mental health) *issues at the moment, so it’s difficult to get him to understand things at the moment…It’s been massively difficult*.” (P11, male, 50, home working, non-smoker, household with children, access to private outdoor space)

These and other parents described how lockdown made it more difficult for their teenage children who smoke to obtain tobacco, and how the relative ease with which they could purchase vaping devices and oils meant e–cigarettes were used by under-age smokers as a temporary substitute for tobacco when confined at home:

“*He* (teenage son) *couldn’t get access to anything and anyone* (to obtain cigarettes during full lockdown)…. *He was ordering, you know, liquid stuff for vapes* (online), *but as soon as that went and he was able to see friends, then the pattern changed, basically* (he went back to smoking)…*I mean, it is illegal for him to smoke, he’s only 15… The vape stuff, they don’t really control very well on websites, but, obviously, tobacco, they do more so.*” (P11, male, 50, home working, non-smoker, household with children, access to private outdoor space)

### 3.4. Diversionary Activity

Some participants identified outdoor activity and gardening in particular as a means of ‘keeping busy’ during lockdown. Such activities were associated with reductions in smoking, with good weather conditions and longer daylight hours contributing to these changes:

“*I think probably during the hot weather, I didn’t smoke as much. I would keep myself occupied doing other things. Like, when I was gardening, I wouldn’t be smoking because I was too busy. So the more…I suppose the busier I was, the less I smoked*.” (P03, male, 71, retired, smoker, all adult household, access to private outdoor space)

Similarly, undertaking internal home improvement projects during lockdown could result in significant reductions in indoor smoking, as householders were keen to instigate new smoke-free rules at home following redecoration. These changes were also facilitated by warm weather conditions reported during the lockdown period, which made smoking outside or by an entrance door easier:

“*Before lockdown I would smoke indoors. But during lockdown, I had to make myself a bit busy, as you do, and I said to my wife, ‘right, let’s decorate the house’. And I hadn’t realised how much the smoke affected the colour of the walls… and that was it, I turned round and said, ‘That’s it, to save me keep painting like this, I’m smoking outside or by the front door’… so it was completely zero* (smoking) *indoors after I’d decorated.*” (P06, male, 55, ft/carer, smoker, household with children, access to private outdoor space)

These displacement effects resulted in reduced smoking in some cases, particularly when newly decorated areas where smoking was no longer permitted were spaces people had previously used to relax and enjoy a cigarette:

“*Now we’ve got to actually get up* (from our seat) *and go into the kitchen and spend, say, five minutes away from what you’re doing so you think, ‘I’ll wait ‘til the end of this’ or whatever* (before I go through to the kitchen for a smoke)*….And also, as well, we’ve got a habit of only having half* (a cigarette) *and then coming back and having the other half a bit later on. Whereas that wouldn’t happen if we were sitting (relaxing having a smoke) in the front room*.” (P14, female, 54, home working, smoker, all adult household, access to private outdoor space)

## 4. Discussion

This study provided novel insights regarding the impacts of changed routines associated with lockdown restrictions on household smoking behaviours. Findings suggest several factors associated with spending more time at home—loss of routine, confinement, and stress, smoking concealment, and diversionary activities—were associated with changes in smoking behaviour. In many cases, working from home was associated with increased smoking and the potential displacement of smoking to indoor home spaces.

Our findings support previous studies suggesting that the Covid-19 pandemic has been associated with bidirectional changes in smoking: it has prompted some smokers to increase and others to decrease their smoking [3,4,5,6,7,8]. Pandemic-related stressors have been suggested to account for increased smoking in some cases during lockdown [4,6,7,8] and our findings support this suggestion. An additional part of the explanation may lie in how the pandemic changed the structure of everyday lives: altering daily routines significantly and confining people largely to the home. Boredom associated with enforced time at home during the pandemic has been associated with more frequent smoking in recent studies [4,8]. However, our study suggests that while the pandemic may have intensified smoking for many smokers, it has also disrupted established patterns of smoking associations and behaviours. Previous research has suggested that smoking behaviours are often strongly bound up in habit and routine, and are reinforced by regular temporal and spatial cues (e.g., smoking during the mid-morning work-break, or whilst waiting at the bus stop) [19,20]. During lockdown, the routines, spaces, and places which previously constituted daily smoking environments were often altered or no longer accessible for some smokers. For example, being away from the workplace or not being able to socialise facilitated a reduction in smoking for some of our interviewees.

Our findings compliment those of recent research conducted in the UK and Italy [8,10], suggesting that some smokers reduced their daily cigarette consumption as a result of changes to their daily smoking environment brought about by lockdown. In other cases, participants spoke of new motivations to create a smoke-free home, after embarking on home improvement projects during lockdown and in relation to the arrival of warmer weather. These findings support those in the wider smoke-free homes literature, where several studies have shown that decorating indoors can facilitate a move to smoking outdoors, and that bad weather can act as a barrier to maintaining a smoke-free home [15].

Recent research has suggested that the pandemic and lockdown may have provided a ‘teachable moment’ that has prompted changes in health behaviours, including smoking [9]. We did not specifically ask participants whether they viewed any changes to their smoking as short-term responses and adaptations to the specific circumstances of lockdown, or as longer-term changes they would continue beyond lockdown. This was a limitation of our study, and research conducted over a longer-time frame would be required to explore this. Most participants in our study found that stresses and boredom associated with the constraints of lockdown contributed to increased smoking. These findings highlight the importance of continued public health messaging on smoking, especially as we emerge from the Covid-19 pandemic. Bidirectional changes in health behaviours associated with lockdown have also been reported in other contexts including alcohol use and vaping [9,21,22]. On this basis, it has been suggested that public health policies, measures and media are required to promote greater self-awareness, self-help, and self-care within the home setting to prevent later strains on the healthcare system [23]. Identifying potential health-related consequences of the Covid-19 pandemic and lockdown measures will assist with developing appropriate public health responses, which is especially important given the pandemic is still ongoing.

Our study found that reductions in smoking and reduced smoking in the home were often relatively passive insofar as they were an unplanned consequence of the disruption to usual daily routines brought about by lockdown. Reducing smoking before quitting may appeal to populations who find it particularly difficult to quit smoking, however studies in these populations are limited [24]. Furthermore, creating a smoke-free home has been suggested to facilitate cessation and increase quit attempts in the 6 months that follows [25]. Research and monitoring are required to explore whether adapting these positive health behaviour changes during lockdown provides a ‘stepping stone’ to cessation in the longer-term, for whom, and under what circumstances. As the UK and other countries look to ‘build back better’ there is an opportunity to develop health improvement messages encouraging consideration of how smoking has changed during lockdown and the potential benefits to health in sustaining positive changes in the longer-term. This messaging style could, for example, draw on the *maintenance* phase of the Comprehensive Messaging Strategy for Sustained Behaviour Change (CMSSBC) [26], which informs recipients on how to persist with a newly adopted health behaviour in the face of potential challenges. Messaging strategies could also be developed to emphasise that disruption to life caused by Covid-19 and lockdown, and other stages of life where usual routines are disrupted (e.g., hospitalisation, changes in employment), can present opportunities for reflecting on smoking, and smoking in the home, with a view to making positive changes. Similarly, the revelation of younger family members smoking to parents due to their confinement together during lockdown highlights the potential for establishing positive dialogues between parents and children regarding smoking.

Our findings suggest that in some cases, home working was associated with increases in smoking and the displacement of smoking to indoor spaces. This finding raises concerns regarding increased SHS exposure for non-smoking family members, including children. Whilst homeworking was on a gradual, but slow, upward trajectory in the UK even before the pandemic, it rose considerably during the first lockdown, from 5.7% of workers in January/February 2020 to 43.1% in April 2020 [27]. The new ways of working associated with Covid-19 and lockdown are likely to have a long-term impact on the nature of working spaces and may limit the extent to which knowledge-based work returns fully to the ‘workplace’ [28]. With recent UK government advice [29] discouraging the use of shared workstations and hot-desks in office spaces, home may continue to feature as a place of work for some time.

In the UK, employers must by law prevent people from smoking at work if within an enclosed or substantially enclosed space. Considerable progress has been made in the UK and elsewhere in reducing the proportion of adult smokers who smoke in the home, including through the introduction of smoke-free public places legislation and SHS media campaigns, which have changed social norms related to SHS exposure and increased understanding of the health hazards to non-smokers and children of exposure to SHS [30,31,32]. We are not aware of current guidance available on the provision of smoke-free support to employees working from home. However, employers have the same occupational health and safety responsibilities for home workers as for any other workers regarding other issues such as exposure to hazardous materials, use of display screen equipment, stress, and mental health. With any transition from Covid-related norms to new working practices, the issue of protection from SHS in the home as a place of work requires further attention.

Our findings highlight the potential ways in which disruption to routines during lockdown have impacted on smoking consumption and home smoking behaviours. Whilst the qualitative approach utilised means our findings are not generalisable, the use of quota sampling methods does ensure multiple perspectives and experiences of lockdown are represented. Conducting telephone interviews at the time of the first full lockdown would have enabled exploration of the impacts of lockdown on smoking behaviours ‘in the moment’, which may be more accurate than retrospective accounts. However, our study was able to develop short case histories to illustrate how smoking behaviour evolved over time, from the beginning of the first full lockdown until the time of the interview, which incorporated the shift to partial lockdown (10 May to 4 July 2020). Whilst social desirability may have led to some under-reporting of increased smoking/smoking in the home (especially in smoking households with children), previous research suggests that the anonymity of telephone interviews (compared with face-to-face methods) may better equip participants to respond honestly and openly to potentially sensitive questions during qualitative interviews [33]. However, ‘non-visual’ methods such as telephone interviews also have potential disadvantages compared to use of remote methods such as Zoom or Skype, which may better assist participants to form and maintain a rapport with researchers [34]. The lack of nonverbal communication using telephone interviews may also contribute to communication barriers [33], although we did not experience these challenges. Nearly two-fifths of our interviews relied on third party accounts to describe changes to smoking behaviours made by other family members. These accounts may have been subject to a greater risk of inaccuracy and/or incompleteness. However, due to the confined conditions in which participants were living it was possible in some instances for participants to consult with other family members regarding their behaviour.

## 5. Conclusions

Whilst changes in levels of smoking during lockdown were bidirectional, many reported changes related to increased smoking were often explained by a shift to working from home. Similar divergence was also observed in changes to smoking at home, with home improvement projects responsible for householders implementing new restrictions on smoking at home and home working responsible for the relaxation/breaking of rules prohibiting smoking indoors. The emergence of new home working norms highlights the need for employers to support staff to maintain smoke-free workplaces during transitions from work-to-home spaces.

## Figures and Tables

**Table 1 ijerph-18-05816-t001:** Participant characteristics (n = 25).

Participant	Sex	Age	Social Grade ^1^	Work Status ^2^	Access to Private Outdoor Space ^3^	Household Type ^4^	Smoke-Free Home ^5^	Household Smoking Status ^6^	Participant Smoking Status	Impact on Smoking ^7^	Impact on no. of Hours Spent at Home Each Day ^8^
P01	Male	71	E	retired	yes	all adult	yes	smoking	non-smoker	no change	+2 h 30 m
P02	Male	66	B	retired	yes	single	no	smoking	smoker	inc- > dec	+2 h 00 m
P03	Male	71	E	retired	yes	all adult	yes	mixed	smoker	no change	+2 h 50 m
P04	Male	45	C2	temp	yes	family	no	smoking	smoker	inc	+8 h 00 m
P05	Male	41	D	work	no	family	yes	smoking	smoker	no change	+1 h 00 m
P06	Male	55	E	ft/carer	yes	family	no	mixed	smoker	dec	+1 h 00 m
P07	Male	31	D	unemp	no	single	yes	smoking	smoker	no change	+10 h 58 m
P08	Female	45	C2	h/home ^9^	yes	all adult	yes	smoking	smoker	dec	−2 h 30 m
P09	Male	40	C2	furl	no	family	yes	mixed	non-smoker	dec	+6 h 24 m
P10	Female	45	C1	furl	no	family^11^	no	Smoking ^13^	smoker	dec- > inc	+8 h 28 m
P11	Male	50	B	h/work	yes	family	yes	Mixed ^14^	non-smoker	not known	+7 h 10 m
P12	Male	61	B	work ^10^	yes	all adult	yes	mixed	smoker	inc	+10 h 00 m
P13	Male	36	C1	h/work	yes	family	yes	mixed	smoker	inc	+9 h 45 m
P14	Female	54	C1	h/work	yes	all adult	no	smoking	smoker	dec	+8 h 42 m
P15	Male	31	D	furl	yes	family	yes	smoking	smoker	inc- > dec ^15^	+6 h 00 m
P16	Male	65	E	retired	yes	all adult	yes	mixed	non-smoker	no change	+1 h 15 m
P17	Female	22	D	furl	no	family^12^	no	mixed	non-smoker	inc	+4 h 40 m
P18	Female	68	C1	retired	no	single	yes	smoking	smoker	no change	+2 h 20 m
P19	Female	54	C1	h/work	no	single	no	smoking	smoker	inc	+11 h 30 m
P20	Female	28	C2	furl	no	family	yes	mixed	non-smoker	dec	+8 h 50 m
P21	Female	28	C2	unemp	no	family^12^	yes	mixed	non-smoker	inc- > dec	+3 h 05 m
P22	Female	73	E	retired	yes	all adult	yes	mixed	non-smoker	no change	+0 h 30 m
P23	Female	70	B	furl	yes	all adult	yes	mixed	non-smoker	inc	+7 h 15 m
P24	Female	33	C1	h/work	no	family^11^	no	smoking	smoker	inc	+8 h 10 m
P25	Female	61	B	h/work	no	all adult	no	mixed	smoker	inc	+4 h 05 m

^1^ Social grade was based on household chief income earner: A—higher managerial, professional; B—intermediate managerial, professional; C1—supervisory or clerical, junior managerial, student; C2—skilled manual worker; D—semi or unskilled manual work, casual worker; E—retired, unemployed, not-working due to long-term sickness, full-time carer, homemaker. ^2^ Participant work status: ‘Ft/carer’—full timer carer; ‘Work’—carried on working as before (see points 10 and 11 for variations); ‘H/work’—required to work from home; ‘Furl’—furloughed; ‘Unemp’—(made) unemployed; ‘Temp’—forced to temporarily cease working. NB: participants work status could change during and following lockdown. Status identified relates to that reported at recruitment. ^3^ Private outdoor spaces included gardens, yards, and private balconies. ^4^ Household types: ‘All adult’—householders all aged 18 or over; ‘single’—households comprising a lone adult; ‘family’—households comprising one or more children aged under 18. ^5^ ‘Smoke-free’ households—householders who describe their homes as smoke-free or as seeking to move all smoking at home to outdoor spaces. ^6^ Household smoking status: ‘smoking’—all adult householders’ were smokers; ‘mixed’—one or more adult householders were smokers (see footnote 14). ^7^ Impact of lockdown on smoking consumption relates to the participants smoking or, where the participant was a non-smoker, to the main smoking member of the household; inc: increased; dec: decreased: inc- > dec: increase followed by a decrease; dec -> inc: decrease followed by an increase; not known: insufficient data. ^8^ Impact on number of hours spent at home relates to the difference between hours spent inside on a typical day before and during the period of full-lockdown. ^9^ P08 increased the number of days she worked from home from three to five days during and following lockdown spending more time outside in her garden. ^10^ P12 moved to home working during the full lockdown then later restarted outdoor site visits whilst continuing to under-take all office-based work from home. ^11^ P10 and P24 were both single parents. ^12^ P17 and P21 were both pregnant during lockdown. ^13^ P10 also identified her teenaged daughter as a smoker. ^14^ P11 identified his teenaged son as the only smoker in this household at interview. ^15^ P15 made a sustained quit attempt after experiencing a dramatic increase in the amount he was smoking during lockdown.

## Data Availability

No additional data is available.

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
