# Peer review of "‘I Was Smoking a Lot More during Lockdown Because I Can’: A Qualitative Study of How UK Smokers Responded to the Covid-19 Lockdown"

_ijerph, 2021, doi:10.3390/ijerph18115816_

Round 1
Reviewer 1 Report
This study explored the changes of smoking patterns during COVID-19 lockdown COVID-19 is still spreading in many parts of the world despite the vaccination efforts and the findings from the study have implications for other parts of the world. Although the paper is very well written, it has shortcomings:
1 There are blank spaces throughout the paper, like the ones in Line 48, 49.
2 While exclusion criteria for recruitment are clear, inclusion ones are not very clear.
3 Whereas 572 responders met the inclusion criteria, why only 25 were eventually invited?
4 Please provide information on the measures to ensure the quality of the study (rigor of the study)
5 One of the study purposes was to explore the family-dynamics-based smoking behaviors. However, some of the participants were in single living style. Please explain why these people were not excluded.
6 “Then the transcripts and coded data for all participants were re-read and analysed separately to build a series of individual narratives and case histories”. It is not clear that after the initial discussion between the researchers, the analysis of the subsequent data was done by one or two researchers or the group of the researchers.
7 The words “across a 24- hour day increased” are vague. Delete “day”?
8 The words “I wouldn't smoke 359 in public, like.”are vague. Delete “like”?
9 There are four themes emerged from the data: Changes to routine and boredom, Confinement and stress, Smoking concealment, and Diversionary activity. While three of the themes were the influencing factors to the changed smoking patterns at home: Changes to routine and boredom, Confinement and stress, and Diversionary activity, the theme “Smoking concealment” was the changed smoking pattern. The themes are not parallel.
10 The lockdown has influenced the smoking patterns of the smokers in bi-direction, as claimed by the researchers. I suggest the researchers change the title of the paper to “A qualitative study of smoking patterns at home in responding to the Covid-19 lockdown”
Author Response
Response to Reviewer 1: Manuscript number 1208929
We are very pleased to be given an opportunity to respond to comments from Reviewer 1. We thank them for their time and for thoughtful comments and appreciate their enthusiasm for the paper and their support for its publication subject to revisions.
We are pleased to provide responses to their suggestions for revisions below. We hope that you will agree that we have been able to respond positively to the helpful comments and guidance, and that you will consider the paper improved and acceptable for publication.
This study explored the changes of smoking patterns during COVID-19 lockdown COVID-19 is still spreading in many parts of the world despite the vaccination efforts and the findings from the study have implications for other parts of the world. Although the paper is very well written, it has shortcomings:
Many thanks for your encouraging comments regarding the relevance and presentation of our paper.
1.1 There are blank spaces throughout the paper, like the ones in Line 48, 49.
Thank you for highlighting this formatting issue. We have checked the latest version of the manuscript for revision carefully, and have not found blank spaces in lines 48 and 49 or in other places. This issue may have been resolved already if the editorial office has made formatting changes to our original submission.
1.2 While exclusion criteria for recruitment are clear, inclusion ones are not very clear.
In lines 98-99, we note that all panel members aged 21 or over were invited by email to take part in the study and complete a screening questionnaire. Being aged 21 or over was the only inclusion criteria applied during recruitment. We have now clarified in lines 100-107 all exclusion criteria that were applied once screening questionnaires were received from potential participants.
1.3 Whereas 572 responders met the inclusion criteria, why only 25 were eventually invited?
In lines 106-115 we note that 32 participants were recruited in order of response to meet a quota sampling strategy based on seven sample variables: gender, age, smoking status, family status, household composition, household access to private outdoor space, and pandemic-related change to work-life status. This paper presents findings from all households with smokers (n=25).
1.4 Please provide information on the measures to ensure the quality of the study (rigor of the study)
The study was reported in accordance with the 32-item checklist of Consolidated Criteria for Reporting Qualitative Research (COREQ). We have included reference to the COREQ reporting guidelines in lines 154-155. We have also uploaded this checklist with our revised submission.
We have added detail of other measures taken to ensure study rigour. We note that our interview questions were developed through several team discussions (lines 125-126), and that interview questions were reviewed by DE and RO after the first five interviews were conducted, with no adjustments required (lines 136-137). We also note that all audio-files were fully transcribed by a professional transcription agency (lines 140-141) and that verbatim quotes are used (line 198).
The use of a quota sample also assisted to ensure study rigour, ensuring that our sample included participants with a range of different and rich insights and experiences as a function of extent of pandemic-related changes to work-life status, household access/no access to outdoor space, smoking status, and household composition (see lines 111-113).We have also clarified (see lines 153-157) that at regular points during the process of data analysis, the two researchers most closely involved in data collection and analysis met with members of the wider research team to discuss identified codes and categories, the interpretation of data and potential new areas of enquiry.
1.5 One of the study purposes was to explore the family-dynamics-based smoking behaviors. However, some of the participants were in single living style. Please explain why these people were not excluded.
The aim of the study was to explore how Covid-19 lockdown ‘stay at home’ restrictions affected daily smoking routines and behaviours, and to investigate changes to pre-established smoking restrictions in the home. We note in lines 130-133 that the impacts of smoking behaviour on family dynamics was one of several areas explored in interviews. However, we were also interested in how pandemic restrictions may have affected the smoking behaviour of those who did not have to consider others in the household and participants living on their own were not excluded on this basis.
1.6 “Then the transcripts and coded data for all participants were re-read and analysed separately to build a series of individual narratives and case histories”. It is not clear that after the initial discussion between the researchers, the analysis of the subsequent data was done by one or two researchers or the group of the researchers.
Thank you for pointing this out. We have now clarified in lines 152-157 that the subsequent data analysis involved two researchers, and that findings were discussed within the wider team at various points during the analysis process to assist in developing an overall interpretation of the data.
1.7 The words “across a 24- hour day increased” are vague. Delete “day”?
Thank you for your suggestion. We have retained the use of this ’24-hour day’ description, as this was the phrasing used to engage participants in discussions about their daily smoking consumption patterns and the amount of time they spent in the home before and during the first full lockdown each day.
1.8 The words “I wouldn't smoke in public, like.”are vague. Delete “like”?
Many thanks for your helpful suggestion, we have deleted the word ‘like’ on this basis (line 371).
1.9 There are four themes emerged from the data: Changes to routine and boredom, Confinement and stress, Smoking concealment, and Diversionary activity. While three of the themes were the influencing factors to the changed smoking patterns at home: Changes to routine and boredom, Confinement and stress, and Diversionary activity, the theme “Smoking concealment” was the changed smoking pattern. The themes are not parallel.
Thank you for raising this point. Our themes reflect key topics and patterns of meaning that came up repeatedly in our data, and as they represent diverse participant experiences, we would not necessarily expect them to run in parallel. However, some of the examples we present to illustrate smoking concealment suggest concealment was also an influencing factor which changed smoking patterns. For example, in lines 373-383, we highlight the case of a mother who in seeking to conceal her smoking from her children to maintain her position in their eyes as a non-smoker, reduced the amount she smoked. We have considered your point carefully, and on that basis we now note in lines 197-198, that the themes identified reflected both potential influencers on smoking behaviour and ways in which smoking behaviours changed.
1.10 The lockdown has influenced the smoking patterns of the smokers in bi-direction, as claimed by the researchers. I suggest the researchers change the title of the paper to “A qualitative study of smoking patterns at home in responding to the Covid-19 lockdown”
We agree that our results do show that lockdown had bidirectional influences on smoking consumption. However, most participants found lockdown increased the amount of time spent at home, where stresses associated with confinement, curtailment of social routines, removal of barriers and distractions to smoking due to home working, and feelings of boredom all contributed to increased smoking. Fewer factors were identified as reducing smoking during lockdown. We have given your suggestion careful consideration, in conjunction with Reviewer 2’s view that the paper title is ‘clear, informative and engaging’. We feel our title reflects the majority of participants’ experiences, and we have retained it on that basis.
Reviewer 2 Report
Reviewer comments
Thank you for the opportunity to review this submission - this is certainly a timely and original contribution and I enjoyed the opportunity to read and review this work. This piece clearly addresses a gap in current literature and represents a welcome addition to what is necessarily only an emerging field of study during the COVID-19 pandemic. The backdrop and context to the study is clearly explained, and methods are generally communicated succinctly and clearly. Findings are explained well, with very effective use of quotations to illuminate points made and bring the article to life. Implications of findings are discussed, although as I suggest below, the value and contribution of this kind of research could have been drawn out a little more explicitly – with more consideration given to the wider applicability of findings (for example how might they link to wider research on other health behaviours, and do participants suggest that changes to smoking habits could be long-term (or not?!) and extend beyond the pandemic?). This article is very well-written and engaging, making it easy for the reader to follow.
I would be delighted to recommend this work for publication and have only minor comments / suggestions for consideration.
General comments
I felt that, on balance, the authors could have done a little more to show / explain why research in this field (including their own) is important and valuable. This is sometimes implicitly raised but could at times be a little more explicit e.g. mention in more detail the points around lockdown more widely as an opportunity to explore changes in people’s health behaviours, routines and habits? How does this kind of research contribute to our wider understandings about, for example, smoking or other health behaviours? Does it have broader value / implications beyond just understanding smoking practices within the very specific (and hopefully short-term!) context of a national lockdown?
Linked into thinking about the wider applicability of your findings, perhaps mention at some point research into other health behaviour changes during lockdown – the obvious comparison is alcohol consumption, where, as you also suggest in terms of smoking, evidence seems quite mixed i.e. not really a uniform increase or decrease in drinking, but possibly a polarisation (heavy drinkers drinking more, lighter drinkers drinking less). For relevant alcohol research, see Garnett, C., Jackson, S.E., Oldham, M., Brown, J., Steptoe, A., & Fancourt, D. (2021). Factors associated with drinking behaviour during COVID-19 social distancing and lockdown among adults in the UK, Drug Alcohol Depend 219:108461 and Oldham, M., Garnett, C., Brown, J., Kale, D., Shahab, L. and Herbec, A. (2021). Characterising the patterns of and factors associated with increased alcohol consumption since COVID‐19 in a UK sample. Drug Alcohol Rev. https://doi.org/10.1111/dar.13256
Did participants speak about whether they imagined any changes to their smoking would be longer-term / continue beyond lockdown, or was there a suggestion that they would ‘go back to normal’ in terms of their smoking post-lockdown? This might be something worth mentioning, particularly as, if participants did see these kinds of changes as longer-term, then this has implications again for the wider applicability of the findings in a post-COVID world.
More specific comments
- Title is clear, informative and engaging
- Page 2, line 94 – it feels like the introduction ends a little abruptly, I would recommend adding a sentence or two of more in-depth signposting to outline what will be covered for the reader in the remaining sections of the article
- When discussing the sample / methods, it would be useful to be a little clearer about the fact that 32 participants were recruited and 25 were from smoking households, with 7 from non-smoking households. This was clear when I looked at the supplementary table but when I read the methodology section as a standalone document I was left feeling a little unsure as to why the other 7 participants hadn’t been included in your discussion / analysis here
- Could you reflect briefly on any possible disadvantages of using telephone interviews as a methodology? You discuss benefits (in that participants may feel able to be more honest) but I wondered if you might also acknowledge the fact that telephone interviews do not allow for observation of facial expressions, body language and other visual cues in a way that might have been possible had platforms such as Zoom or Skype been used
Author Response
Response to Reviewer 2: Manuscript number 1208929
We are very pleased to be given an opportunity to respond to Reviewer 1’s comments. We thank the Reviewer for their time and for thoughtful comments and appreciate their enthusiasm for the paper and their support for its publication subject to revisions.
We are pleased to provide responses to their suggestions for revisions below. We hope that you will agree that we have been able to respond positively to the helpful comments and guidance, and that you will consider the paper improved and acceptable for publication.
Thank you for the opportunity to review this submission - this is certainly a timely and original contribution and I enjoyed the opportunity to read and review this work. This piece clearly addresses a gap in current literature and represents a welcome addition to what is necessarily only an emerging field of study during the COVID-19 pandemic. The backdrop and context to the study is clearly explained, and methods are generally communicated succinctly and clearly. Findings are explained well, with very effective use of quotations to illuminate points made and bring the article to life. Implications of findings are discussed, although as I suggest below, the value and contribution of this kind of research could have been drawn out a little more explicitly – with more consideration given to the wider applicability of findings (for example how might they link to wider research on other health behaviours, and do participants suggest that changes to smoking habits could be long-term (or not?!) and extend beyond the pandemic?). This article is very well-written and engaging, making it easy for the reader to follow.
I would be delighted to recommend this work for publication and have only minor comments / suggestions for consideration.
Many thanks for your encouraging comments regarding the relevance and presentation of our paper.
General comments
2.1. I felt that, on balance, the authors could have done a little more to show / explain why research in this field (including their own) is important and valuable. This is sometimes implicitly raised but could at times be a little more explicit e.g. mention in more detail the points around lockdown more widely as an opportunity to explore changes in people’s health behaviours, routines and habits? How does this kind of research contribute to our wider understandings about, for example, smoking or other health behaviours? Does it have broader value / implications beyond just understanding smoking practices within the very specific (and hopefully short-term!) context of a national lockdown?
Thank you for your helpful suggestion. We have added text in lines 482-496 to discuss these points, noting that it is not clear to what extent changes to smoking practices during the specific and unusual context of a national lockdown may translate into longer term impacts, and that future monitoring and research are needed to clarify whether health behaviour changes made during lockdown are sustained in the longer term. We also refer to research suggesting that some smokers may find it easier to reduce smoking in the run up to cessation rather than stopping smoking abruptly, and that creating a smoke-free home may in turn increase the likelihood of cessation in the longer term. We note also on this basis that research is also required to ascertain whether making these smoking behaviour changes during lockdown provide a ‘stepping stone’ to cessation in the longer-term, for whom, and under what circumstances. We also highlight the potential learning for other major life events and stages of life where usual routines are disrupted (e.g. hospitalisation, changes in employment), which can present opportunities for reflecting on smoking, and smoking in the home, with a view to making positive change (lines 512-518).
2.2 Linked into thinking about the wider applicability of your findings, perhaps mention at some point research into other health behaviour changes during lockdown – the obvious comparison is alcohol consumption, where, as you also suggest in terms of smoking, evidence seems quite mixed i.e. not really a uniform increase or decrease in drinking, but possibly a polarisation (heavy drinkers drinking more, lighter drinkers drinking less). For relevant alcohol research, see Garnett, C., Jackson, S.E., Oldham, M., Brown, J., Steptoe, A., & Fancourt, D. (2021). Factors associated with drinking behaviour during COVID-19 social distancing and lockdown among adults in the UK, Drug Alcohol Depend 219:108461 and Oldham, M., Garnett, C., Brown, J., Kale, D., Shahab, L. and Herbec, A. (2021). Characterising the patterns of and factors associated with increased alcohol consumption since COVID‐19 in a UK sample. Drug Alcohol Rev. https://doi.org/10.1111/dar.13256
Thank you for raising this point about wider applicability and research into other health behaviours during lockdown. We have added text in lines 489-496 to incorporate this discussion point. We note that bidirectional changes in alcohol use and vaping have also been reported (amongst others). We highlight the importance of identifying potential health-related consequences of the Covid-19 pandemic and lockdown measures to assist with developing appropriate public health responses, and how important this is given the pandemic is still ongoing.
2.3 Did participants speak about whether they imagined any changes to their smoking would be longer-term / continue beyond lockdown, or was there a suggestion that they would ‘go back to normal’ in terms of their smoking post-lockdown? This might be something worth mentioning, particularly as, if participants did see these kinds of changes as longer-term, then this has implications again for the wider applicability of the findings in a post-COVID world.
This is an important point and a limitation of our study, as we did not specifically ask all participants about this (and nor did they spontaneously offer this information). In cases where we relied on third parties to comment on changes in smoking they had observed amongst other family members, we were not well positioned to seek this information. This was a limitation of our study, and an area for future research which we now note in lines 482-486. Generally speaking, most participants who made a positive change to their smoking didn’t seem to view this as a long term opportunity, and in most cases changes were relatively passive (see lines 497-498). On this basis, we highlight the importance of health improvement messages encouraging consideration of how smoking has changed during lockdown and the potential benefits to health in sustaining positive changes in the longer-term (lines 505-518).
More specific comments
2.4 Title is clear, informative and engaging
Many thanks, we are pleased that you feel the title is clear, informative and engaging.
2.5 Page 2, line 94 – it feels like the introduction ends a little abruptly, I would recommend adding a sentence or two of more in-depth signposting to outline what will be covered for the reader in the remaining sections of the article.
We have structured the Introduction in accordance with the IJERPH ‘Instructions for Authors’ guidelines, which include guidance to briefly mention the main aim of the work at the end of the Introduction. On this basis, and to maintain a reasonable word length whilst incorporating other suggestions at this stage, we have refrained from adding additional sentences to signpost content in the remaining sections of the paper. We have however joined the sentence which outlines our aims to the previous paragraph of the introduction to improve the flow of this section of text.
2.6 When discussing the sample / methods, it would be useful to be a little clearer about the fact that 32 participants were recruited and 25 were from smoking households, with 7 from non-smoking households. This was clear when I looked at the supplementary table but when I read the methodology section as a standalone document I was left feeling a little unsure as to why the other 7 participants hadn’t been included in your discussion / analysis here.
We agree with your point that this could have been clearer. We have now clarified in lines 113-115 that seven of the 32 participants were from non-smoking households, alongside our text outlining that this paper presents findings from all households with smokers (n=25).
2.7 Could you reflect briefly on any possible disadvantages of using telephone interviews as a methodology? You discuss benefits (in that participants may feel able to be more honest) but I wondered if you might also acknowledge the fact that telephone interviews do not allow for observation of facial expressions, body language and other visual cues in a way that might have been possible had platforms such as Zoom or Skype been used.
Thank you for your helpful suggestion. We have now included discussion of the potential challenges associated with use of telephone interviews compared to other remote methods such as Skype or Zoom, in lines 550-555.
Round 2
Reviewer 1 Report
The authors have done a great job revising the paper.